# Circular RNAs Involved in the Regulation of the Age-Related Pathways

**DOI:** 10.3390/ijms231810443

**Published:** 2022-09-09

**Authors:** Siqi Wang, Feng Xiao, Jiamei Li, Xiaolan Fan, Zhi He, Taiming Yan, Mingyao Yang, Deying Yang

**Affiliations:** 1College of Animal Science and Technology, Sichuan Agricultural University, Chengdu 611130, China; 2Farm Animal Genetic Resources Exploration and Innovation Key Laboratory of Sichuan Province, Sichuan Agricultural University, Chengdu 611130, China

**Keywords:** circRNAs, aging, age-related pathway, regulatory roles

## Abstract

Circular RNAs (circRNAs) are a class of covalently circular noncoding RNAs that have been extensively studied in recent years. Aging is a process related to functional decline that is regulated by signal transduction. An increasing number of studies suggest that circRNAs can regulate aging and multiple age-related diseases through their involvement in age-related signaling pathways. CircRNAs perform several biological functions, such as acting as miRNA sponges, directly interacting with proteins, and regulating transcription and translation to proteins or peptides. Herein, we summarize research progress on the biological functions of circRNAs in seven main age-related signaling pathways, namely, the insulin-insulin-like, PI3K-AKT, mTOR, AMPK, FOXO, p53, and NF-κB signaling pathways. In these pathways, circRNAs mainly function as miRNA sponges. In this review, we suggest that circRNAs are widely involved in the regulation of the main age-related pathways and are potential biomarkers for aging and age-related diseases.

## 1. Introduction

Aging is an irreversible process accompanied by various declining functions in organisms, weakened homeostasis, and reduced resistance to environmental damage. It is the largest risk factor that leads to age-related diseases, such as cancer, diabetes, cardiovascular disease, and Alzheimer’s disease [1]. Similar to many other biological processes, aging is regulated by classic signaling pathways and transcription factors. In recent years, research on aging has shifted from simply observing the lifespan to exploring a series of complex intracellular signaling pathways and biological processes [2,3,4].

Genes are an important factor affecting the aging process [5]. Previous studies have found that noncoding RNAs (ncRNAs), including long ncRNAs (lncRNAs), circular RNAs (circRNAs), and microRNAs (miRNAs), play vital regulatory roles in aging and age-related diseases [6,7]. CircRNAs are formed as covalently closed circular structures without 5′ caps or 3′ poly(A) tails. Because of their unique circular structure, circRNAs have more resistance to RNA exonucleases and are more stable than linear RNAs [8]. In 1976, circRNAs were first discovered in viroids [9]. However, because of their low expression abundance, circRNAs are considered to be byproducts of mRNA splicing and have not been further researched [10]. Recent studies suggested that circRNAs are abundant in humans, mice, worms, flies, and other organisms [11], which indicates that circRNAs might be a class of molecules with special functions. Moreover, circRNAs have been implicated in aging, such as showing an age-related accumulation in male *Macaque* brains [12], in mouse cortexes and hippocampi [13], in *C. elegans* [14], and in *Drosophila* heads and photoreceptor neurons [15]. Furthermore, the overexpression of an insulin-sensitive circRNA sulfateless (circSfl) has been confirmed to significantly extend lifespan in *Drosophila* [16].

This article highlights the biogenesis and mechanisms of circRNAs in seven main age-related pathways. We aim to lay a solid foundation for the exploration of circRNAs as biomarkers for the diagnosis and treatment of aging and age-related diseases.

## 2. Biogenesis and Functional Mechanisms of circRNAs

### 2.1. Classification of circRNAs

According to the different biogenesis mechanisms, circRNAs can be divided into exonic circRNAs (ecircRNAs), which participate in only exon circularization; intronic circRNAs (ciRNAs), which participate in only intron circularization; exonic–intronic circRNAs (EIciRNAs), which participate in circularization involving exons and introns; and tRNA intronic circRNAs (tricRNAs), which are derived from splicing pre-tRNA introns [17]. There are three main modes of the formation of ecircRNAs and EIciRNAs: exon-skipping or lariat-driven circularization, direct back-splicing or intron-pairing-driven circularization, and RNA-binding-protein-driven circularization [18]. The biosynthesis of ciRNAs was thought to be initiated by the lariat of removed introns during the pre-mRNA splicing process and mainly depended on a 7-nt GU-rich element near the 5′ splice site and an 11-nt C-rich element close to the branch-point site [19]. The formation of tricRNA requires tRNA splicing enzymes to divide pre-tRNA into two parts: tricRNAs are generated by a 3′–5′ phosphodiester bond, and the other part generates tRNAs [20].

### 2.2. Functional Mechanisms of circRNAs

CircRNAs regulate biological processes mainly through four molecular functions. At present, the most extensive research on the functional mechanisms of circRNAs has been carried out with respect to their actions as molecular sponges of miRNAs, and there are multiple miRNA response elements on the circular sequences [21]. One of the most well-studied circRNAs is cerebellar degeneration-related protein 1 antisense RNA (CDR1as), which contains more than 70 binding sites of miR-7 in its sequence and is also known as a circRNA sponge for miR-7 (ciRS-7) or CDR1NAT [22]. CircRNAs can inhibit miRNA activity by adsorption and affect the expression of downstream target genes [22]. Furthermore, the same circRNA might target different miRNA–mRNA axes to perform diverse functions in different diseases [23,24]. However, some research has demonstrated that most circRNAs contain fewer miRNA binding sites and do not possess the properties of effective miRNA sponges [25].

In addition to functioning as miRNA sponges, some circRNAs can also directly interact with RNA-binding proteins (RBPs), such as circRNA muscleblind (circMbl) which could adsorb MBL protein to maintain dynamic stability of circular and linear RNAs [26]; or regulate gene transcription [27], such as circRNA ras homolog family member T1 (circRHOT1) which recruited TIP60 to the NR2F6 promoter and initiated NR2F6 transcription [28]. In addition, recent research has shown that circRNAs, which are modified with internal ribosome entry sites (IRESs) [29] or N6-methyladenosine (m6A) [30], might have the potential to translate in a cap-independent way to perform biological functions. For example, circMbl encoded a protein of about 6.5 Kda which was modulated by starvation and FOXO [31] (Figure 1).

## 3. Age-Related Signaling Pathways

Functional decline in aging and age-related diseases is associated with several highly conserved cell signaling pathways, which form a complex network through multiple molecular reciprocal regulations and regulate lifespan and age-related diseases. Based on previous studies, we selected and described seven classic and well-researched longevity signaling pathways.

### 3.1. Insulin/Insulin-like Growth Factor-1 Signaling (IIS) Pathway

The insulin/insulin-like growth factor-1 (IGF-1) signaling pathway was the first established aging pathway [32] (Figure 2a). More than twenty years ago, the mutation of *daf-2*, the gene homologous to the human IGF-1 receptor (IGF-1R), resulted in a twofold extension of the lifespan of *C. elegans* [33], and the effect required DAF-16, the FOXO ortholog in *C. elegans* [34]. The lifespan-extension effect of the decreased activity of insulin receptors is conserved in *Drosophila* [35]. In mammals, reduced growth hormone (GH), insulin, and IGF-1 signaling due to various mutations have also been shown to be associated with longevity phenotypes [36].

Insulin/insulin-like peptide (ILP) binds to insulin receptors on the surface of target cells and activates insulin receptor substrates (IRSs) to initiate an intracellular kinase cascade that culminates in the activation of the kinase AKT. The activation of AKT phosphorylates the downstream transcription factor FOXO, which inhibits the transcriptional function of FOXO, further promoting cell survival, growth, and proliferation [37].

IRSs are components of the downstream IIS pathway. In mammals, IRS1-mutant female mice presented a longer lifespan and various symptoms of delayed aging [38]. The homozygous or heterozygous deletion of IRS2 in the mouse brain also resulted in an increased lifespan and features consistent with delayed aging and/or the attenuation of age-related functional alterations [39]. In *C. elegans*, IRS proteins have not been identified [40], while *Drosophila* expresses a single IRS ortholog protein called chico. The mutation of chico has been proven to significantly extend the *Drosophila* median lifespan by up to 48% in homozygotes and 36% in heterozygotes [41].

### 3.2. PI3K/Akt Signaling Pathway

Phosphatidylinositol 3-kinase (PI3K) is an intracellular kinase that acts as a key molecule in the initiation of signal transduction pathways after the binding of extracellular signals to cell surface receptors that, together with mTOR, appear to play a role in aging and lifespan [42] (Figure 2b). Protein kinase B (AKT, also known as PKB) is a major effector during PI3K-driven cell signal transduction in response to extracellular stimuli, and AKT activity is upregulated by PI3K signaling during the activation of receptor tyrosine kinases or G-protein-coupled receptors [43].

PI3K is activated by multiple cell surface receptors and forms PIP3 on the cell membrane. PIP3 is a secondary messenger that activates downstream proteins, the most important of which is the phosphoinositide-dependent protein kinase-1 (PDK1), which controls the activation of AKT signal transduction [44]. PIP3 binds AKT and PDK1 and promotes the phosphorylation of AKT at Thr308. However, AKT activation also requires its phosphorylation at Ser473 by mTORC2 [45,46]. Activated AKT affects downstream targets, including GSK3, p21, FOXO, and mTOR, thus regulating multiple intracellular signaling pathways that affect cell growth, proliferation, differentiation, apoptosis, migration, secretion, angiogenesis, transcription, and protein synthesis [47,48].

### 3.3. mTOR Signaling Pathway

The mechanistic target of rapamycin (mTOR) is an evolutionarily conserved serine-threonine kinase that belongs to the PI3K-related kinase family and forms part of two structurally and functionally distinct complexes, mTORC1 and mTORC2 (Figure 2c). mTORC1 contains mTOR, mammalian lethal with sec-13 protein 8 (mLST8), DEP domain containing mTOR-interacting protein (Deptor), the Tti1/Tel2 complex, regulatory-associated protein of mammalian target of rapamycin (Raptor), and proline-rich Akt substrate 40 kDa (PRAS40). mTORC1 is inhibited by rapamycin, integrates diverse environmental and intracellular signals, such as growth factors and nutrients, and subsequently regulates diverse cellular processes, including metabolism, survival, growth, differentiation, and autophagy [49]. mTORC2 is composed of mTOR, mLST8, Deptor, the Tti1/Tel2 complex, rapamycin-insensitive companion of mTOR (rictor), mammalian stress-activated map kinase-interacting protein 1 (mSin1), and protein observed with rictor (protor). mTORC2 regulates cellular processes such as metabolism, survival, apoptosis, growth, and proliferation by directly activating Akt, controls ion transport and growth by directly activating serum- and glucocorticoid-induced protein kinase 1 (SGK1), and regulates cell shape in a cell-type-specific fashion by affecting the actin cytoskeleton by activating protein kinase C-α (PKC-α) [50].

The inhibition of mTOR signaling delayed aging and extended the lifespan in yeast [51], *Drosophila* [52], *C. elegans* [53] and mice [54]. Two key downstream effectors of mTORC1, ribosomal protein S6 kinase (S6K) and eukaryotic translation initiation factor 4E-binding protein (4E-BP), regulate mRNA translation, ribosome biogenesis, and protein synthesis [3]. Both decreased S6K activity and increased 4E-BP activity could extend the lifespans of multiple species.

### 3.4. AMPK Signaling Pathway

AMP-activated kinase (AMPK) is a conserved serine/threonine kinase consisting of a catalytic α subunit and two regulatory β and γ subunits that plays a fundamental role in energy metabolism in cells and organisms [55] (Figure 2d). AMPK signaling activation decreases with aging and may be associated with many age-related diseases, such as cardiovascular disease and metabolic syndrome. The overexpression of AMPK (AMPK ortholog named AAK-2 in *C. elegans*) extended the lifespans of *C. elegans* and *Drosophila* [56].

An increased level of AMP/ADP activates AMPK, thereby reducing the utilization of ATP by inhibiting the synthesis of glycogen, protein, and cholesterol and inducing the production of ATP by promoting fatty acid oxidation and glycolysis [57]. AMPK is also activated by upstream kinases, including Ca^2+^/calmodulin-dependent protein kinase β (CaMKKβ), serine/threonine kinase 11 (LKB1), and transforming growth factor-β-activated kinase 1 (TAK1), by phosphorylating the catalytic α subunit at Thr172. However, activated AMPK can be inhibited by protein phosphatases (PPs), such as PP2A, PP2Cα, and Ppm1E [58]. AMPK controls a complex signaling network with other longevity pathways. AMPK affects autophagy via the mTOR pathways, enhancing resistance against stress by the FOXO/DAF-16 and sirtuin 1 (SIRT1) pathways and inhibiting the inflammatory response by suppressing NF-κB signaling [59]. It has been stated that lifespan extension induced by AMPK is mediated by CRTC-1 and CREB signaling [60].

### 3.5. FOXO Signaling Pathway

Forkhead box O (FOXO) represents a subfamily of the Forkhead family of transcription factors and is conserved from *C. elegans* to mammals (Figure 2e). There is only one FOXO gene in the invertebrate genome (daf-16 in worms and dFOXO in flies), while there are four in mammals, namely, FOXO1, FOXO3, FOXO4, and FOXO6. FOXO proteins mainly act as transcriptional activators that bind to the consensus core recognition motif TTGTTTAC and are inhibited by the IIS pathway [61]. The activated IIS pathway triggers the PI3K-Akt pathway and then allows AKT to phosphorylate FOXO factors at three conserved residues. The phosphorylation of FOXO leads to its exit from nuclei and transport to the cytoplasm, resulting in the suppression of the FOXO-dependent transcription of target genes. Conversely, in the absence of IIS, FOXO is translocated into the nucleus and activates the expression of FOXO-dependent target genes [62]. In addition, other kinases, such as AMPK, JNK, and ERK, can also phosphorylate FOXO. In addition to phosphorylation, some diverse posttranslational modifications, including acetylation, deacetylation, methylation, and ubiquitination, have also been shown to modify the subcellular localization, protein levels, DNA binding, and transcriptional activity of FOXO factors [63].

In flies, dFOXO overexpression significantly extended the lifespan [64]. In *C. elegans*, lifespan extension was mediated by a loss-of-function mutation of Daf-2/IGF-1, which is associated with Daf-16, the ortholog of FOXO in worms [65]. There are several mechanisms by which FOXO promotes longevity, such as participating in autophagy, improving cellular antioxidant capacity, and maintaining stem cell homeostasis [61].

### 3.6. p53 Signaling Pathway

p53 is a central effector of many stress-related molecular cascades, and plays a crucial role in tumor suppression and aging by regulating DNA repair, cell cycle progression, cell death, and senescence [66] (Figure 2f). p53 regulates cellular senescence via the activation of the cyclin-dependent kinase inhibitor 1 CDKN1A (p21) and promyelocytic leukemia protein (PML) [67]. Some of the effects of p53 on organismal aging are mediated by autophagy [68] directed at the IIS and mTOR pathways or through MDM2 [69]. MDM2 is the major negative regulator of p53 and interacts with p53 to form a stable complex. On the one hand, MDM2 binds to p53 to prevent the transcriptional activation of p53 and promote p53 degradation through ubiquitination. On the other hand, p53 stimulates MDM2 transcription by binding to its promoter region. Therefore, there is a regulatory feedback loop between p53 and MDM2 [70].

### 3.7. NF-κB Signaling Pathway

The nuclear factor kappa-light-chain-enhancer of the activated B cells (NF-κB) protein family complex contains five different transcription factors: p50, p52, RelA (p65), RelB, and c-Rel [71] (Figure 2g). In the cytoplasm, NF-κB exists in an inhibited state and is sequestered by a series of NF-κB inhibitors (i.e., IκB). Exposure to one of several inducing stimuli results in the phosphorylation of IκB proteins by the IκB kinase (IKK) complex, followed by ubiquitination, the proteasomal degradation of IκB-inhibiting NF-κB, and the concomitant translocation of NFκB to the nucleus, where it functions as a gene transcriptional regulator of key biological processes. In the nucleus, NF-κB, together with transcriptional coactivators, fine-tunes the activity of RNA polymerase (Pol) II at different stages of the transcription cycle [72].

NF-κB plays an important role in cell proliferation, apoptosis, immunity, and inflammation, and can respond to oxidative stress, DNA damage, immune activation, and growth regulatory signals [73,74]. During the aging process, NF-κB activity is significantly increased, and it has been reported that NF-κB activity is repressed by several longevity genes, such as *Daf-16/FOXO3a* [75] and sirtuin 1 (*SIRT1*) [76].

## 4. CircRNAs in Age-Related Signaling Pathways

### 4.1. IIS Pathway

In the IIS pathway, circRNAs mainly act as miRNA sponges to regulate the expression level of IGF-1R or IRS2/4, for instance, CDR1as/miR-7 [77], circRUNX1/miR-145-5p [78], circ_0067835/miR-296-5p [79], circ_0014130/miR-142-5p [80], circ_0000517/miR-326 [81], circPLK1/miR-4500 [82], circPDHX/miR-378a-3p [83], hsa_circ_0002577/miR-625-5p [84], hsa_circ_0020850/miR-195-4p [85], hsa_circ_0023409/miR-542-3p [86], circ_0000003/miR-338-3p [87], and circFAM126A/miR-613 [88] (Figure 3). In this review, three circRNAs are discussed.

#### 4.1.1. CDR1as

CDR1as is a specific circRNA with multiple miRNAs binding sites on its sequence, and it has been confirmed that it increase with age in mouse cortex samples [14]. The dysregulation of CDR1as may lead to a series of aging-related diseases, including Alzheimer’s disease [89], diabetes [90], and various cancers [91,92], which suggests its grander prospects in the field of pathologic diagnosis and targeted therapy. CDR1as is more highly expressed in colorectal cancer (CRC) tissues than in normal tissues. Silencing CDR1as inhibited the proliferation and invasion of CRC cell lines, and mechanistically, it was dependent on increasing miR-7 expression and suppressing EGFR and IGF-1R expression and further affecting downstream factors. It has also been suggested that CDR1as could serve as a potential molecular target for CRC therapy design [77].

#### 4.1.2. Hsa_circ_0020850

Hsa_circ_0020850 was the most differentially expressed circRNA between normal tissues and lung adenocarcinoma tissues, and was upregulated in the latter. Silencing hsa_circ_0020850 suppressed tumor development, while knocking down miR-195-5p reversed this effect. Dual-luciferase reporter assays revealed that miR-195-5p was targeted via hsa_circ_0020850. Similarly, IRS2 was identified as a downstream target of miR-195-5p, and overexpressing miR-195-4p inhibited tumor development by decreasing IRS2 expression [85]. In total, the knockdown of hsa_circ_0020850 suppressed lung adenocarcinoma cell proliferation, migration and invasion, and facilitated apoptosis, which was mediated by decreasing IRS2 expression by sponging miR-195-4p. In addition, hsa_circ_0020850 might be considered a new target for the treatment of lung adenocarcinoma.

#### 4.1.3. Hsa_circ_0023409

The expression levels of hsa_circ_0023409 were higher in gastric cancer (GC) tissues than in adjacent normal tissues. GC patients with high hsa_circ_0023409 expression showed poorer survival rates than those with low hsa_circ_0023409 expression. Hsa_circ_0023409 overexpression promoted the growth, migration, and invasion of GC cells by functioning as a miR-542-3p sponge. Furthermore, the targeting relationship between miR-542-3p and IRS4 was identified by dual-luciferase reporter assays. When hsa_circ_0023409 was overexpressed, the protein expression level of IRS4 was increased, and this effect could be reversed by miR-542-3p and vice versa. The increased expression of IRS4 further activated the downstream PI3K/AKT pathway [86]. In summary, hsa_circ_0023409 promotes GC progression by sponging miR-542-3p and elevating the expression of IRS4, thus activating the downstream PI3K/AKT pathway.

### 4.2. PI3K/AKT Signaling Pathway

Numerous studies have shown that the dysregulation or mutation of the PI3K/Akt pathway is one of the most frequent reasons for some age-related diseases, such as various cancers [93]. Herein, we discuss the regulatory roles of circRNAs in multiple cancers through the PI3K/AKT signaling pathway. Most circRNAs function by acting as competing endogenous RNAs (ceRNAs) (Table 1), which indicates that circRNAs have great potential as cancer diagnostic and treatment biomarkers.

#### 4.2.1. Pancreatic Cancer

Pancreatic cancer is a common digestive tract cancer that ranks fourth in cancer-associated mortality around the world. CircRNA eukaryotic translation initiation factor 6 (circEIF6) is significantly increased in pancreatic tumor tissues/cells compared with normal tissues/cells. CircEIF6 promotes pancreatic cancer development by interacting with and decreasing the expression level of miR-557. The downregulation of miR-557 activates the PI3K/Akt signaling pathway by targeting the solute carrier family 7 member 11 (*SLC7A11*) mRNA [94]. The circEIF6/miR-557/*SLC7A11*/PI3K/AKT signaling axis might provide novel therapeutic targets for pancreatic cancer.

Pancreatic ductal adenocarcinoma (PDAC) is the most common subtype of pancreatic cancer, accounting for approximately 85% of all pancreatic cancer cases [118]. PDAC has a very poor prognosis, and the five-year survival rate for pancreatic cancer patients is less than 5%. CircNFIB1 is formed from exons 16 to 18 of the nuclear factor I B (*NFIB*) gene, and it is differentially downregulated in PDAC tissues. CircNFIB1 acts as a miR-486-5p sponge and antagonizes the miR-486-5p-mediated suppression of PIK3R1, which is a regulatory subunit of PI3K, and inhibits the activation of the PI3K/AKT signaling pathway. The inactivation of the PI3K/AKT signaling pathway further suppresses vascular endothelial growth factor-C (VEGF-C) and ultimately suppresses lymphangiogenesis and lymphatic metastasis in PDAC [95].

Another circRNA, circBFAR, is derived from exon 2 of the bifunctional apoptosis regulator (*BFAR*) gene with a length of 336 nt, and is highly expressed in PDAC tissues. CircBFAR promotes the progression of PDAC by sponging miR-34b-5p and upregulating mesenchymal-epithelial transition factor (MET) expression levels. MET overexpression activates downstream Akt-Ser473 phosphorylation and further activates the PI3K/AKT signaling pathway [96]. CircNFIB1 and circBFAR might be potential biomarkers and therapeutic targets for PDAC therapies.

#### 4.2.2. Glioma

Glioma is a prevailing fatal malignancy of the central nervous system that lacks specific treatment targets, and the age-related alterations in neural progenitor cells (NPCs) contribute to both decreased regenerative capacity in the brain and an increased risk of glioma tumorigenesis [119,120]. CircPIP5K1A originates from the phosphatidylinositol-4-phosphate 5-kinase type 1 alpha (*PIP5K1A*) gene, which is highly expressed in glioma tissues compared with normal adjacent tissues. Functionally, it could promote the progression of glioma by elevating the expression level of transcription factor 12 (TCF12) by sponging miR-515-5p, thereby activating the PI3K/AKT pathway [97]. In addition to circPIP5K1A, circ_0000215 [98], circ_0037655 [99], and hsa_circ_0014359 [100] also promote glioma progression by acting as miRNA sponges by activating the PI3K/AKT signaling pathway.

One more-aggressive type of glioma is glioblastoma (GBM), which can occur at any age, but tends to occur more often in older adults. According to extensive investigations, 88% of GBM patients die from the disease within 3 years. GBM remains one of the most challenging malignancies worldwide [121]. Circ-AKT3 is cyclized from exon 3 to exon 7 of the *AKT3* gene and is expressed at lower levels in GBM tissues than in adjacent normal brain tissues. Circ-AKT3 encodes a 174 amino acid novel protein named AKT3-174aa, which is overexpressed and inhibits the GBM malignant phenotype. Importantly, AKT3-174aa, but not circ-AKT3, could function as a tumor suppressor. Mechanistically, AKT3-174aa inhibits GBM tumorigenicity by competitively interacting with phosphorylated 3-phosphoinositide-dependent protein kinase-1 (PDK1) and reducing the phosphorylation of AKT at Thr308, thus negatively regulating the PI3K/AKT signaling pathway [101]. AKT3-174aa, encoded by circ-AKT3, is a potential prognostic marker for GBM patients, and might have future potential clinical uses.

#### 4.2.3. Gastric Cancer

Gastric cancer (GC), one of the most common malignant tumors worldwide, has a low rate of early diagnosis because of lacking and non-specific symptoms [122]. CircRAB31 is derived from exon 2 to exon 5 of the Ras-related protein Rab-31 (*RAB31*) gene, which is downregulated in GC tissues and cells compared with normal tissues and cells. CircRAB31 overexpression inhibits GC proliferation and metastasis in vitro and in vivo, whereas silencing circRAB31 has the opposite effect. Mechanistically, circRAB31 suppresses GC progression by acting as a miR-885-5p sponge and targeting its downstream target, the phosphatase and tensin homolog (PTEN), to further inactive PI3K/AKT signaling [104].

Additionally, the previously discussed circPIP5K1A that promoted the progression of glioma could also promote GC development by binding to miR-671-5p to activate the PI3K/AKT pathway [102]. CircRNA mannosidase alpha class 2B member 2 (circMAN2B2) regulates miR-145 [103], while the mechanism of hsa_circRNA_100269 remains unclear [105], though it is known to modulate GC development through the PI3K/AKT pathway. In summary, circRAB31, circPIP5K1A, circMAN2B2, and hsa_circRNA_100269 might be potential targets for the diagnosis and treatment of GC.

#### 4.2.4. Non-Small Cell Lung Cancer

Over the past two decades, important advances have been made in the treatment of non-small cell lung cancer (NSCLC); however, the overall cure and survival rates of NSCLC remain low [123]. We herein discuss three circRNAs, circFARSA, circ-PLCD1, and circ-PITX1, which might be promising biomarkers for the diagnosis and treatment of NSCLC.

CircFARSA is 338 nucleotides long and includes exons 5–7 of the phenylalanyl-tRNA synthetase subunit alpha (*FARSA*) mRNA, which is highly expressed in NSCLC tissues and cells compared with normal tissues and cells. Functionally, circFARSA promotes NSCLC progression and macrophage differentiation. Mechanistically, circFARSA accelerates macrophage polarization to the immunosuppressive M2 phenotype by promoting the ubiquitination and degradation of PTEN and activating the PI3K/AKT pathway to accelerate NSCLC metastasis. In addition, circFARSA could be combined with eukaryotic translation initiation factor 4A3 (EIF4A3), which promotes circRNA biogenesis and cyclization, at the flanking sequences to mediate circRNA circularization and expression in NSCLC cells [106].

Based on the results of circRNA high-throughput sequencing in NSCLC tissues and normal tissues, circ-PLCD1, which is circularized from exon 14 to exon 15 of the phospholipase C delta 1 (*PLCD1*) gene, was found to have the largest differential expression. Circ-PLCD1 is significantly downregulated in NSCLC tissues and cell lines, and the overexpression of circ-PLCD1 inhibits the malignant phenotype of NSCLC cells. Mechanistically, circ-PLCD1 acts as a sponge to interact with miR-375 and miR-1179 and elevate PTEN expression to suppress PI3K/AKT signaling, thereby repressing NSCLC tumorigenesis [107].

CircRNA-paired like homeodomain 1 (circ-PITX1) is substantially upregulated in NSCLC tissues and cells. Functionally, the overexpression of circ-PITX1 promotes NSCLC development, whereas its silencing results in the opposite effect. Similar to circ-PLCD1, circ-PITX1 facilitates NSCLC proliferation and metastasis by sponging miR-30E-5p, which then targets the 3′ untranslated region (UTR) of integrin subunit alpha 6 (ITGA6) and ultimately activates the PI3K/AKT pathway [108].

#### 4.2.5. Colorectal Cancer

Colorectal cancer (CRC) is the third most common cause of cancer-related death throughout the world, and originates as a result of alterations in the normal colon or rectum epitheliums [124]. CircPTEN is derived from the line mRNA PTEN, which is notably expressed at low levels in CRC tissues and cells. Upregulated circPTEN inhibits CRC cell proliferation, migration, and invasion, whereas silencing circPTEN results in the opposite effect. Mechanistically, on the one hand, circPTEN sponges miR-4470 and elevates PTEN expression to suppress AKT; on the other hand, circPTEN competitively interacts with the ring-finger domain of TNF receptor-associated factor 6 (TRAF6) to inhibit K63-linked AKT ubiquitination and AKT phosphorylation at Thr308 and Ser473 [109]. Overall, circPTEN modifies CRC progression by regulating AKT through two molecular functions, which provides new insights for CRC therapies.

Hsa_circ_0008285 is derived from exon 2 of the chromodomain Y like (*CDYL*) gene, and has a mature sequence length of 667 nucleotides. Circ_0008285 is downregulated in CRC tissues and cells and is associated with CRC growth and metastasis. Circ_0008285 acts as a CRC suppressor by interacting with miR-382-5p to elevate PTEN expression, which inactivates PI3K/Akt signaling [110].

Another circRNA, circCDYL2, generated from exon 2 of the *CDYL2* gene, was found based on high-throughput sequencing data, and is upregulated in a highly migratory CRC cell subline. CircCDYL2 increases CRC cell migration in vitro by binding to the Ezrin protein, which is a cytoskeletal organizer that promotes tumor metastasis by reorganizing the cytoskeleton or controlling signal transduction and preventing its degradation to enhance AKT phosphorylation [111]. circRNAs originating from both the *CDYL* and *CDYL2* genes have been indicated as potential therapeutic targets for CRC treatment.

#### 4.2.6. Breast Cancer

Breast cancer (BC) is the most common cancer leading to mortality among females worldwide. It has been determined that hsa_circ_001569 is upregulated in both BC tissues and cells. Previous studies have demonstrated that hsa_circ_001569 plays functional roles in osteosarcoma (OS) [125], HCC [126], and CRC [127] by acting as a miRNA sponge. In BC, silencing hsa_circ_001569 results in the suppression of BC cell growth and metastatic potential, and this effect is due to the impediment of the PI3K-AKT signaling pathway. However, the mechanism by which hsa_circ_001569 modulates the PI3K-AKT pathway needs to be further studied [112]. Hsa_circ_001569 might have potential as a target for BC therapy.

#### 4.2.7. Other Cancers

CircRNA electron-transfer flavoprotein subunit alpha (circETFA) promotes HCC development by upregulating the C-C motif chemokine ligand 5 (CCL5) expression level to further regulate the PI3K/AKT signaling pathway and other key downstream effectors. The mechanisms involve sponging hsa-miR-612 to block the inhibitory role of hsa-miR-612 on CCL5 on the one hand and recruiting EIF4A3 to prolong the half-life of CCL5 mRNA on the other hand [113]. In addition, circIGF1R regulates PI3K/p-AKT levels to promote the progression of HCC, while the detailed mechanism by which circIGF1R affects the PI3K/AKT pathway requires further investigation [114].

The mechanisms are similar to those of circETFA; circRNA-9119 suppresses ovarian cancer (OC) cell viability via the miR-21-5p/PTEN axis [115], and circRNA coiled-body phosphoprotein 1 (circNOLC1) promotes prostate cancer (Pca) cell proliferation and migration in vitro and in vivo through the miR-647/progestin and adipoQ receptor family member 4 (PAQR4) axis [116]. In bladder cancer, circZNF139 promotes cell proliferation, migration, and invasion by activating the PI3K/AKT pathway, but the detailed mechanisms are still unknown [117].

### 4.3. mTOR Signaling Pathway

mTOR is a central factor in the signal transduction network, and can be activated by AKT but is inhibited by AMPK. We divided the circRNAs involved in the mTOR signaling pathway into three parts, namely, those involved in the AKT-mTOR axis, those involved in the AMPK-mTOR axis, and those that directly target mTOR. Among them, circRNAs mainly function as miRNA sponges.

#### 4.3.1. AKT-mTOR Axis

mTOR is extensively corroborated as a crucial downstream molecule of AKT, and numerous studies have indicated that circRNAs can regulate age-related diseases through the AKT-mTOR axis. CircHIPK3 is derived from exons 7–11 of homeodomain-interacting protein kinase 3 (*HIPK3*), which is highly expressed in lung cancer tissues and cells. CircHIPK3 knockdown suppresses lung cancer cell proliferation, migration, and glycolysis, while facilitating apoptosis. This effect is mediated by blocking the AKT-mTOR axis via targeting miR-381-3p [128].

Similarly, hsa_circ0001666 [129], circRNA membrane bound O-acyltransferase domain containing 2 (circMBOAT2) [130], and circRNA nuclear factor of activated T cells 3 (circNFATC3) [131] could also act as ceRNAs to regulate tumor progression via the AKT-mTOR axis (Figure 3). Some circRNAs, such as hsa_circ_0079929 [132], function as tumor regulatory factors through the Akt-mTOR axis, but the detailed mechanisms are unclear.

#### 4.3.2. AMPK-mTOR Axis

mTOR activity can be inhibited by AMPK. We herein discuss two circRNAs, circRNA_002581 and circWHSC1, that act as miRNA sponges to regulate age-related diseases through the AMPK-mTOR axis. In nonalcoholic steatohepatitis (NASH), circRNA_002581 acts as a miR-122 sponge and then upregulates the expression of its target gene, cytoplasmic polyadenylation element-binding protein 1 (*CPEB1*), which subsequently impairs autophagy via the PTEN–AMPK–mTOR regulatory pathway, thereby exacerbating NASH progression. Conversely, the antagonizing circRNA_002581 shows the opposite effect [133]. Another circRNA derived from the wolf-hirschhorn syndrome candidate gene-1 (*WHSC1*) gene, circWHSC1, is highly expressed in BC tissues. The overexpression of circWHSC1 promotes BC development and boosts xenograft tumor growth in nude mice, which is mediated by sponging miR-195-5p. FASN is considered a target of miR-195-5p, and can modulate the downstream AMPK-mTOR pathway. Overall, circWHSC1 expedites BC progression by acting as a miR-195-5p sponge to target FASN and inactivate AMPK while activating mTOR [134].

#### 4.3.3. Targeting mTOR

In addition to cocrossing with other signaling pathways, circRNAs can directly target the mTOR signaling pathway, mainly by acting as miRNA sponges. CircMYLK is derived from exons 25–29 of myosin light chain kinase (*MYLK*), and has been reported to be an oncogenic factor in several cancer types. Silencing circMYLK inhibits cervical cancer (CC) cell growth, and this effect is mediated by impaired mTORC signaling. Mechanistically, circMYLK sponges miR-1301-3p elevate the expression of Ras homolog enriched in brain (RHEB), which is an essential upstream modulator of mTOR signaling activity, and further results in mTOR signaling activation and the CC cell malignant phenotype [135].

Similar to circMYLK, circRNA-100338 [136], hsa_circ_0011324 [137], and hsa_circ_0037251 [138] could also function by acting as miRNA sponges to target mTOR signaling (Figure 3).

### 4.4. AMPK Signaling Pathway

In addition to the above-discussed circRNAs functioning through the AMPK-mTOR axis, we observed that circACC1, which is derived from exons 2–4 of the human acetyl-CoA carboxylase 1 (*ACC1*) gene, could regulate CRC progression by activating AMPK. CircACC1 plays a critical role in cellular responses to metabolic stress, and has been reported to function as a tumor regulator in GC [139] and NSCLC [140] by acting as a ceRNA. In CRC, circACC1 silencing or overexpression results in growth inhibition or promotion, respectively. Mechanistically, circACC1 binds to the regulatory β and γ subunits of AMPK and forms a ternary complex to stabilize and promote AMPK holoenzyme activity. The activation of AMPK mediated by circACC1 regulates glycolysis and fatty acid β-oxidation in cells, and might play a pathological role in CRC [141].

### 4.5. p53 Signaling Pathway

In the p53 signaling pathway, we summarize three circRNAs, circ-PGAP3 [142], circ_0021977 [143], and circ_100395 [144], that function as miRNA sponges; three circRNAs, CDR1as [145], circ-Sirt1 [146], and circSCAP [147], that directly interact with proteins (Figure 3); and one circRNA, circ-MDM2 [148], that does not have a defined mechanism.

Among them, CDR1as has been discussed before, and functions as a CRC promoter by binding to miR-7 in the IIS pathway, while in the p53 signaling pathway, it can directly bind to the DNA-binding domain (DBD) region of p53 to restrict its interaction with MDM2 and prevent p53 degradation to inhibit GBM [145].

#### 4.5.1. Circ-PGAP3

Circ-PGAP3 is derived from the post-GPI attachment to the proteins phospholipase 3 (*PGAP3*) gene. In CC tissues and cells, circ-PGAP3 is significantly downregulated. The overexpression of circ-PGAP3 improves the poor prognosis of CC and significantly inhibits cell proliferation in vitro and tumor growth in vivo. This tumor-suppressive effect of circ-PGAP3 is mediated by sponging miR-769-5p and further increasing the expression levels of p53 and its downstream targets [142].

#### 4.5.2. Circ-Sirt1

Circ-Sirt1 is derived from the well-known longevity gene *SIRT1*, and was previously reported to inhibit GC development by sponging miR-132-3p/miR-212-3p [149] and to inhibit VSMC proliferation by regulating the oncogene c-Myc [150]. Furthermore, the activated circ-Sirt1/SIRT1 axis has been confirmed to function in a manner which inhibits oxidative stress and inflammation both in vivo and in vitro [151].

Vascular smooth muscle cells (VSMCs) are the major components of the blood vessel wall, and are closely associated with age-related vascular diseases. It was demonstrated that VSMC senescence could promote neointima formation by increasing intimal migration, oxidative stress, inflammation, and collagen deposition following vascular injury [152]. Circ-Sirt1 is highly expressed in young and healthy arteries, but is downregulated in the aged arteries and neointima of humans and mice. The overexpression of circ-Sirt1 delays Ang II-induced VSMC senescence in vitro and ameliorates neointimal hyperplasia in vivo. Mechanistically, circ-Sirt1 decelerates VSMC senescence and ameliorates neointimal formation by repressing p53 activity, not only by binding to and blocking p53 nuclear translocation but also by promoting SIRT1-mediated p53 deacetylation and inactivation [146].

#### 4.5.3. CircSCAP

CircSCAP is derived from exons 3–5 of the SREBF chaperone (*SCAP*) gene, and is significantly downregulated in lung cancer tissues and negatively associated with poor prognosis. In vitro, circSCAP inhibits proliferation and migration but promotes apoptosis in NSCLC, while in vivo, the ectopic expression of circSCAP suppresses tumor growth. CircSCAP interacts with the splicing factor 3a subunit 3 (SF3A3) protein and facilitates the degradation of SF3A3 by promoting its ubiquitination. SF3A3 directly binds to protein arginine methyltransferase 5 (PRMT5), and the degradation of SF3A3 weakens the formation of the SF3A3/PRMT5 complex, enhances the expression level of MDM4-S, and further activates downstream p53 signaling to inhibit the malignancy of NSCLC [147].

#### 4.5.4. Circ-MDM2

Based on the sequencing data of CRC cell lines (HCT116, RKO and SW48) that were untreated or treated with a DNA-damaging agent, circ-MDM2 was selected as the target circRNA, the expression of which was altered upon DNA damage and dependent on p53. Circ-MDM2 is formed from exons 4–8 of the p53-inducible gene *MDM2*, and is upregulated after DNA damage treatment. Silencing circ-MDM2 impairs CRC growth in vivo, and this effect is dependent on p53. However, the molecular mechanisms that govern the circ-MDM2/p53 axis remain to be thoroughly investigated [148].

### 4.6. NF-κB Signaling Pathway

Circ-TPGS2 is generated from the tubulin polyglutamylase complex subunit 2 (*TPGS2*) gene, which is upregulated in metastatic BC tissues compared with nonmetastatic tissues. The overexpression of circ-TPGS2 promotes BC cell migration, while silencing circ-TPGS2 results in the opposite effect. Mechanistically, circ-TPGS2 sponges miR-7 and elevates TRAF6 expression levels, resulting in p65 phosphorylation and nuclear translocation, ultimately dysregulating the tumor microenvironment and promoting BC cell-motility by activating NF-κB signaling. Moreover, p65 activates circ-TPGS2 transcription, forming a positive feedback loop and amplifying the prometastatic effect of circ-TPGS2 [153]. Similar to circ-TPGS2, circRNA GLIS family zinc finger 2 (circGLIS2) [154] also function by acting as miRNA sponges through the NF-κB pathway.

In addition to acting as miRNA sponges, circRNAs can also function by interacting with proteins. CircCORO1C is generated from coronin 1C (*CORO1C*), which is significantly upregulated in HCC. Silencing circCORO1C inhibits the tumorigenesis of HCC cells in vivo and in vitro, while overexpressing circCORO1C leads to proliferation and metastasis. Mechanistically, circCORO1C activates the NF-κB signaling pathway, promotes P65 phosphorylation, and upregulates c-Myc and COX-2, further leading to increased programmed death-ligand 1 (PD-L1) expression and ultimately regulating HCC progression [155]. In addition, circRNA *cMras* could inhibit lung adenocarcinoma progression by interacting with alpha-beta hydrolase domain 5 (ABHD5) and adipose triglyceride lipase (ATGL) through the NF-κB signaling pathway [156].

## 5. Summary and Prospects

Aging is a complex process, with gradual degenerative changes in the body increasing the risk of occurrence of aging-related diseases, such as cancers, diabetes, autoimmune diseases, infections, and cardiovascular and cerebrovascular diseases. Similar to many other biological processes, the aging process is also regulated by canonical signaling pathways and transcription factors, including the IIS pathway, PI3K/Akt pathway, AMPK pathway, mTOR pathway, FOXO pathway, p53 pathway, and NF-κB pathway. CircRNAs are a class of circular noncoding RNAs without a 5′- cap or 3′- poly(A) tail that can act as miRNA sponges, bind with proteins, regulate transcription, and/or directly translate proteins to exert their biological functions. In recent years, numerous studies have shown that circRNAs are differentially expressed in various tumor tissues/cells compared with normal tissues/cells and play a regulatory role in diverse age-related diseases.

In this review, we described seven classic age-related pathways, focusing on the research progress of circRNAs in these pathways. Numerous studies have suggested that circRNAs play a regulatory role in aging and age-related diseases via these pathways, which indicates that circRNAs might have the potential to become diagnostic and therapeutic biomarkers for age-related diseases. However, research on circRNAs in the FOXO signaling pathway is lacking. With the advancement of biotechnology, this pathway may soon be better understood.

## Figures and Tables

**Figure 1 ijms-23-10443-f001:**
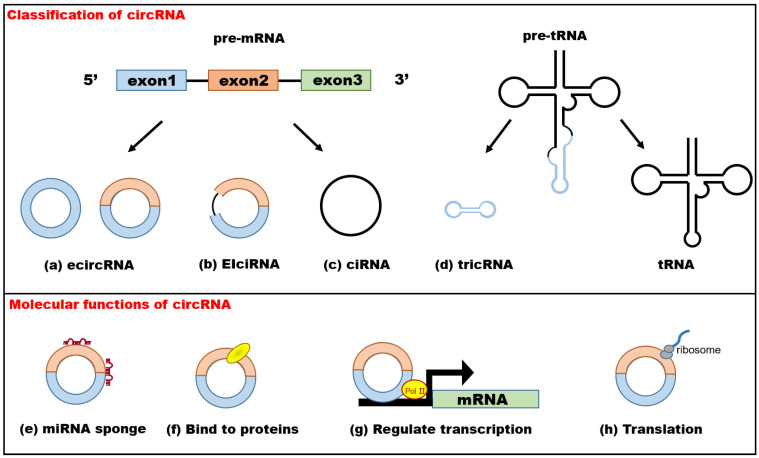
Classification and molecular functions of circRNAs. The top of the figure shows that circRNAs are divided into 4 categories: (**a**) exonic circRNAs (ecircRNAs), (**b**) exon–intron circRNAs (EIciRNAs), (**c**) intronic circRNAs (ciRNAs), and (**d**) tRNA intronic circular RNAs (tricRNAs). The bottom of the figure shows four potential functions of circRNAs: (**e**) microRNA (miRNA) sponging: some circRNAs serve as efficient miRNA sponges, regulating the activity of miRNA target genes; (**f**) binding to proteins: circRNAs affect protein function directly; (**g**,**h**) regulation: some circRNAs regulate transcription and encode peptides or proteins if they have internal ribosome entry sites (IRESs).

**Figure 2 ijms-23-10443-f002:**
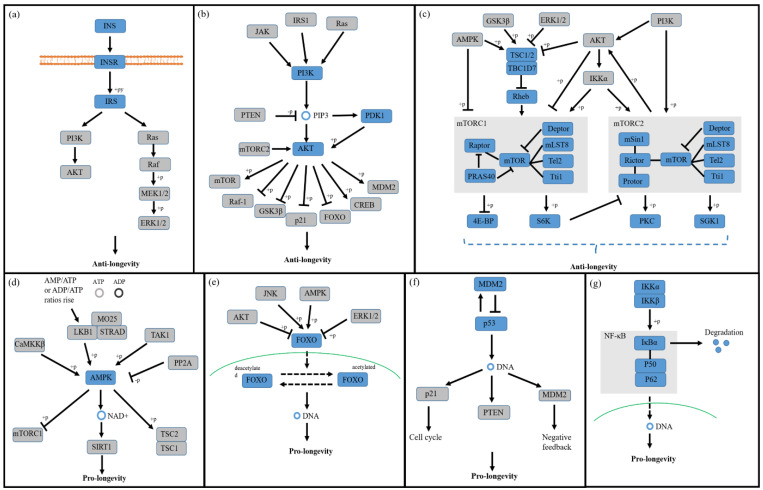
Signal transduction through seven age−related signaling pathways. (**a**) IIS signaling pathway; (**b**) PI3K−AKT signaling pathway; (**c**) mTOR signaling pathway; (**d**) AMPK signaling pathway; (**e**) FOXO signaling pathway; (**f**) p53 signaling pathway; (**g**) NF−κB signaling pathway. The blue squares indicate the core factors in each pathway, while the gray squares indicate the upstream or downstream targets.

**Figure 3 ijms-23-10443-f003:**
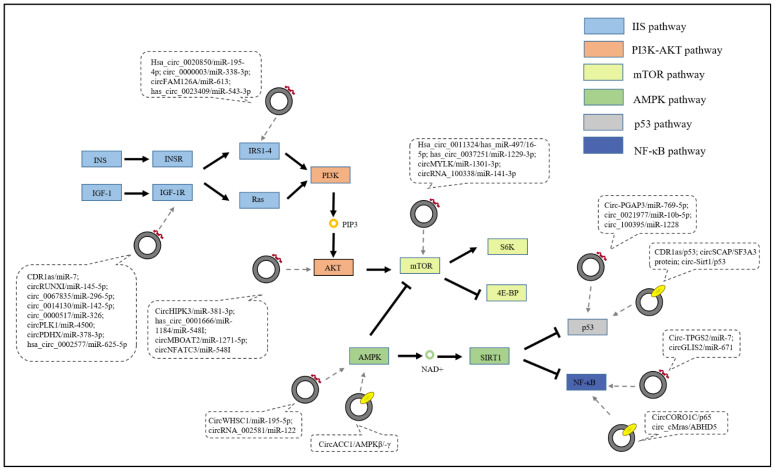
An overview of circRNAs involved in the regulation of age-related signaling pathways. CircRNA/miRNA indicates circRNAs that act as miRNA sponges, circRNA/protein indicates circRNAs that directly bind to proteins, and circRNA/mRNA indicates circRNAs that regulate the transcription of mRNA.

**Table 1 ijms-23-10443-t001:** CircRNAs that regulate age-related diseases via the PI3K/Akt signaling pathway.

Disease	CircRNA	Functional Mechanism	Targets	References
Pancreatic cancer	CircEIF6	miRNA sponge	miR-557/*SLC7A11*	[94]
PDAC	CircNFIB1	miRNA sponge	miR-486-5p/PIK3R1	[95]
PDAC	CircBFAR	miRNA sponge	miR-34b-5p/MET	[96]
Glioma	CircPIP5K1A	miRNA sponge	miR-515-5p/TCF12	[97]
Glioma	Circ_0000215	miRNA sponge	miR-495-3p/CXCR2	[98]
Glioma	Circ_0037655	miRNA sponge	miR-214	[99]
Glioma	Hsa-circ-0014359	miRNA sponge	miR-153	[100]
GBM	Circ-AKT3	translation to protein	PDK1	[101]
GC	CircPIP5K1A	miRNA sponge	miR-671-5p	[102]
GC	CircMAN2B2	miRNA sponge	miR-145	[103]
GC	CircRAB31	miRNA sponge	miR-885-5p	[104]
GC	Hsa_circRNA_100269	unclear		[105]
NSCLC	CircFARSA	interact with protein	PTEN	[106]
NSCLC	Circ-PLCD1	miRNA sponge	miR-375/miR-1179/PTEN	[107]
NSCLC	Circ-PITX1	miRNA sponge	miR-30E-5p/ITGA6	[108]
CRC	CircPTEN	miRNA sponge	miR-4470/PTEN	[109]
CRC	Circ_0008285	miRNA sponge	miR-382-5p/PTEN	[110]
CRC	CircCDYL2	interact with protein	Ezrin	[111]
BC	Hsa_circ_001569	unclear		[112]
HCC	CircETFA	miRNA sponge	hsa-miR-612/CCL5	[113]
HCC	CircIGF1R	unclear		[114]
OC	CircRNA-9119	miRNA sponge	miR-21-5p/PTEN	[115]
PCa	CircNOLC1	miRNA sponge	miR-647/PAQR4	[116]
Bladder cancer	CircZNF139	unclear		[117]

Note: PDAC, pancreatic ductal adenocarcinoma; GBM, glioblastoma; GC, gastric cancer; NSCLC, non-small cell lung cancer; CRC, colorectal cancer; BC, breast cancer; HCC, hepatocellular carcinoma; OC, ovarian cancer; PCa, prostate cancer.

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
