# Peer review of "Circular RNAs Involved in the Regulation of the Age-Related Pathways"

_ijms, 2022, doi:10.3390/ijms231810443_

Round 1

Reviewer 1 Report

Circular RNAs play roles in several age-related pathway. Specific circular RNAs could serve as biomarkers for aging and age-related diseases.  The interest in circular RNAs has surged since 2015. The growth in the number of papers has been exponential. 481 papers were published on this topic in 2021, according to a search in PubMed.

There have been eight systematic reviews on circular RNAs. They have generally summarized the roles of circular RNAs in specific cancers. None of these reviews address the role of circular RNAs in aging.

The authors try to fill this gap by reviewing the role of circular RNA in seven age-related pathways. The authors briefly review the biogenesis, classification, and functional mechanisms of circular RNAs.

Then the authors review seven age-related signaling pathways. After laying this foundation of knowledge, the authors review the role of circular RNAs in each pathway. The results of many studies suggested that circular RNAs play a role in the age-related pathway. They may be potential therapeutic targets. This is a timely and authoritative review of this topic.

The manuscript is well written.  I found only one thaw: the journal titles in the bibliography need to be reformatted. They are presently in sentence case.

Author Response

Response to Reviewer 1 Comments

Point 1: The journal titles in the bibliography need to be reformatted. They are presently in sentence case.

Response 1: Thanks for your suggestion. The journal titles in the bibliography have been reformatted.

Reviewer 2 Report

In current review, authors summarized some circRNAs’ function in different pathways, which include the insulin-insulin-like, PI3K-AKT, mTOR, AMPK, FOXO, p53, 19 and NF-κB signaling pathways. In these pathways, circRNAs mainly function as miRNA sponges. It is a great review, however, there are some comments as below should be modified. Authors should prepare a major revision for publish in second review.

Aging is named a long time-dependent biological process that could accumulate many changes during the time at a lot of physiological processes, which lead to or associate with diseases and death. There are numbers of hallmarks that could indicate some common denominators of aging, for example, genomic instability, epigenetic changes, and proteostasis dysregulation. Age is the important risk factor for multiple diseases, which includes neurodegenerative disease, chronic non-communicable diseases. Likewise, circRNAs have been implicated in age, such as enriching in various tissues of aged rodents, in rhesus macaque brain, during aging in Caenorhabdities elegans, and increasing with age in Drosophila.

1.     However, in this review, they only discussed the circRNAs in different pathways function and lacked important details to describe the aged-related circRNA involved into these pathways. authors should change title and aims for current review. For example, circCDR1as, its function in AD could as age-related circRNA. But authors listed CDR1as-miR-7 axis in islet cells to regulate insulin transcription and secretion, didn’t illustrated the function in age. All circRNAs, which listed in current paper, were not discussed their function in aged-related disease or model.

2.     Authors showed some diseases in Table 1 and described that were age-related disease. As we know, glioma is most common in older (over 65) and children (under 12). But in reference 89, the published didn’t demonstrate circPIP5K1A played its role in age process or aging. Thus, please authors check all circRNAs and the diseases in paper whether are related to age.

3.     In line 64-77, authors listed four functions of circRNAs, could authors give some examples of circRNAs?

Authors should think about their review’s conclusion, maybe not discuss circRNA has relation with age.

Author Response

Response to Reviewer 2 Comments

Point 1: However, in this review, they only discussed the circRNAs in different pathways function and lacked important details to describe the aged-related circRNA involved into these pathways. authors should change title and aims for current review. For example, circCDR1as, its function in AD could as age-related circRNA. But authors listed CDR1as-miR-7 axis in islet cells to regulate insulin transcription and secretion, didn’t illustrated the function in age. All circRNAs, which listed in current paper, were not discussed their function in aged-related disease or model.

Response 1: Thanks for your constructive suggestions. We have added the expression pattern of circRNAs in the aging process (lines 43-47), and discussed the relationship between circRNAs, such as CDR1as and circSirt1, and aging-related diseases (lines 258-262 and lines 528-530). However, there are few studies on the function of other circRNAs in aging-related diseases or models, except for the diseases that we have discussed.

Point 2: Authors showed some diseases in Table 1 and described that were age-related disease. As we know, glioma is most common in older (over 65) and children (under 12). But in reference 89, the published didn’t demonstrate circPIP5K1A played its role in age process or aging. Thus, please authors check all circRNAs and the diseases in paper whether are related to age.

Response 2: We have checked all the diseases in our manuscript carefully. Then, we found that the triple-negative breast cancer (TNBC), osteosarcoma (OS), oral squamous cell carcinoma (OSCC), esophageal squamous cell carcinoma (ESCC), laryngeal squamous cell carcinoma (LSCC) and nasopharyngeal carcinoma (NPC) may not be related to aging, which have been removed from the revised manuscript.

And for the glioma mentioned by the reviewer, the research has shown that the age-related alterations in neural progenitor cells (NPCs) lead to both decreased regenerative capacity in the brain and an increased risk of tumorigenesis, particularly the most common adult-onset brain tumor, glioma. (Attach the references: [1] Stoll EA, Horner PJ, Rostomily RC. The impact of age on oncogenic potential: tumor-initiating cells and the brain microenvironment. Aging Cell. 2013 Oct;12(5):733-41. [2] Mikheev AM, Ramakrishna R, Stoll EA, Mikheeva SA, Beyer RP, Plotnik DA, Schwartz JL, Rockhill JK, Silber JR, Born DE, Kosai Y, Horner PJ, Rostomily RC. Increased age of transformed mouse neural progenitor/stem cells recapitulates age-dependent clinical features of human glioma malignancy. Aging Cell. 2012 Dec;11(6):1027-35.)

Point 3: In line 64-77, authors listed four functions of circRNAs, could authors give some examples of circRNAs?

Response 3: Thanks for your constructive suggestions. We have added four examples corresponding to the four functions (lines 71-74 and lines 80-89). Since this part is not the focus of our paper, it is briefly described.

Point 4: Authors should think about their review’s conclusion, maybe not discuss circRNA has relation with age.

Response 4: We have deleted the content of circRNAs in the diseases which were not aging-related, so we do not consider changing the conclusions of our review.

Round 2

Reviewer 2 Report

Authors responded all comments from first version, the paper could be published on IJMS.